# Combination of acalabrutinib with lenalidomide and rituximab in relapsed/refractory aggressive B-cell non-Hodgkin lymphoma: a single-arm phase II trial

Changhee Park [1], Ho Sup Lee[2], Ka-Won Kang [3], Won-Sik Lee[4], Young Rok Do[5], Jae-Yong Kwak [6], Ho-Jin Shin[7], Sung-Yong Kim[8], Jun Ho Yi[9], Sung-Nam Lim[10], Jeong-Ok Lee[11], Deok-Hwan Yang [12], Hun Jang[13], Byoungsan Choi[13], Jiwoo Lim[14], Choong Hyun Sun[14], Ja Min Byun [1], Sung-Soo Yoon [1] & Youngil Koh [1,14] ✉

Potential synergism between Bruton's tyrosine kinase (BTK) inhibitor and lenalidomide in treating aggressive B-cell lymphoma has been suggested. Here, the authors report a single-arm phase II clinical trial of combination of acalabrutinib, lenalidomide and rituximab (R2A) in patients with aggressive relapsed/refractory aggressive (R/R) B-cell non-Hodgkin lymphoma (NHL). The primary endpoint of this study is objective response rate (ORR), and the secondary endpoints are complete remission (CR) rate, duration of response (DoR), progression-free survival (PFS) and overall survival (OS). A total of 66 patients are enrolled mostly with diffuse large B-cell lymphoma. The ORR is 54.5% and CR rate is 31.8% meeting the primary end point. The median DoR is 12.9 months, and 1-year PFS and OS rate is 33.1% and 67.5% respectively. Adverse events (AE) are manageable with the most frequent AE being neutropenia (31.8%). Patients with *MYD88* mutations, subtypes known for NF-κB activation, and high BTK expression by immunohistochemistry respond well. Overall, these results show a significant efficacy of the R2A regimen in patients with aggressive R/R B-cell NHL, with exploratory biomarkers suggesting potential associations with response. (ClinicalTrials.gov 51 identifier: NCT04094142)

Since the advent of Rituximab, treatment for CD20 positive aggressive B cell non-Hodgkin's lymphoma (NHL) represented by diffuse large B cell lymphoma (DLBCL) has significantly advanced, achieving complete remission (CR) for as high as 76% of patients with first-line rituximab combined chemoimmunotherapy[1,2]. However, as much as one-third of patients experience relapse of the devastating disease[3]. Moreover, around 15% of patients are reported to be refractory to commonly used first line rituximab-based regimens[4]. Although the development of therapeutics such as chimeric antigen receptor (CAR)

T-cell therapy and bispecific T-cell engaging antibody (BiTE) has successfully treated some of relapsed/refractory (R/R) DLBCL patients[5,6], the challenges posed by the high costs and reimbursement issues associated with these therapies have prompted the exploration of alternative approaches[7].

Other promising therapeutic approaches for patients include Bruton tyrosine kinase (BTK) inhibitors, or lenalidomide as monotherapy or in combination[5,6]. Among these, BTK play important roles in B-cell malignancies for cancer cell proliferation and survival. Although

BTK inhibitors have shown promising efficacy and have become the standard of care for treating chronic lymphocytic leukemia, mantle cell lymphoma, and marginal zone B-cell lymphoma as single agents[5], the efficacy of BTK inhibitor monotherapy for aggressive large B-cell lymphoma has not been satisfactory. In a previous trial of ibrutinib monotherapy for DLBCL, the objective response rate (ORR) was 37%, and the duration of response (DoR) was only 4.8 months[8]. Therefore, BTK inhibitors may require an alternative approach for treating aggressive B-cell lymphoma. In this context, a combination strategy with immunomodulatory agents or antibodies that evoke an immune reaction is a reasonable approach based on the effect of BTK on the tumor microenvironment. In fact, in vitro data have shown the potential of BTK inhibition to augment the cytotoxic effect of lenalidomide[9]. Considering the efficacy of antibody-dependent cellular cytotoxicity and complement-dependent cytotoxicity mediated by antibodies, using a BTK inhibitor with an antibody agent would be an attractive combination for aggressive B-cell lymphoma[10].

Theoretically, the combination of a BTK inhibitor with rituximab and lenalidomide may be an effective regimen for R/R B-cell NHL. A trial evaluating the combination of ibrutinib, rituximab, and lenalidomide for patients with R/R DLBCL showed promising efficacy, with an ORR of 44%[11] but also with considerable toxicity, resulting in several patients discontinuing the treatment. In contrast, acalabrutinib, a second-generation BTK inhibitor, could be an alternative part of the combination to minimize toxicity, as it shows improved selectivity for BTK compared to ibrutinib[12].

With advanced molecular classification of DLBCL, BTK pathway has been emphasized in "MCD" or "C5" genetic subtypes associated with *MYD88*/*CD79B* mutations involved in BTK signaling[13,14]. Accordingly, these molecular subgroups may be associated with response to the combination regimen. From the perspective of precision medicine, it is important to discover the molecular subgroups that are potentially responsive to this combination. For example, a previous clinical trial on ibrutinib reported an ORR of 37% in activated B-cell-like DLBCL subjects, among which the ORR was higher in patients with B-cell receptor pathway mutations (55.5%) and even higher in patients with *MYD88* mutations (80%)[8]. Therefore, exploring biomarkers for monitoring the response to the combination regimen is important to design further clinical trials.

In this prospective, open-label, single-arm phase II study, we assess the efficacy and safety of a combination regimen of acalabrutinib with rituximab and lenalidomide (R2A regimen) in patients with R/R B-cell NHL. We also explore potential biomarkers associated with response and used them to identify patients who are likely to respond to the R2A regimen.

## Results
### Patient characteristics
A total of 68 patients were screened at 13 centers in the Republic of Korea from July 11th, 2019, to January 26th, 2021 (Supplementary Fig. 1). Two patients were excluded because of elevated creatinine levels and withdrawal of consent. Finally, 66 patients from 12 centers were enrolled and received the drugs. Patient characteristics are summarized in Table 1. All patients received at least one line of treatment previously with rituximab combination regimen. Four patients previously underwent autologous stem cell transplantation. One patient had previously received an ibrutinib-based combination regimen. None of patient had previously received prior CAR T-cell therapy, which was due to the regulatory and reimbursement issues in Republic of Korea at the time of trial.

Bcl-2 expression data was available for 57 patients, and 47 (82.5%) patients showed Bcl-2 overexpression. Bcl-6 expression data was available for 59 patients had available, and 43 (72.9%) patients exhibited Bcl-6 overexpression. Myc expression data was available for 43 patients, 19 (44.2%) of whom showed Myc overexpression. There were

17 patients having a double-expressor phenotype (both Bcl-2 and Myc overexpression) of DLBCL.

None of the first three patients who received a reduced lenalidomide dose of 15 mg daily in the first cycle experienced hematologic toxicity during the first cycle. Therefore, they received a lenalidomide dose of 20 mg daily in the subsequent cycle. The patients subsequently enrolled received lenalidomide 20 mg daily starting from the first cycle.

### Objective response
The ORR was 54.5% (36 patients, 95% confidence interval [CI] 42.4–66.4). The CR rate was 31.8% (21 patients, 95% CI 21.4–43.9). The ORR in patients with non-germinal center B-cell (nGCB) type DLBCL was 61.7% (95% CI 46.8–74.8) and that in patients with germinal center B-cell (GCB) type DLBCL was 36.4% (95% CI 13.5–66.8). The difference was not statistically significant ($p = 0.179$). The CR rate in patients with non-GCB type DLBCL was 36.2% (95% CI 22.8–51.2) and that in patients with GCB type DLBCL was 18.2% (95% CI 3.3–50.0), and the difference was not statistically significant ($p = 0.310$). Among the other types of lymphoma, one of three patients with DLBCL not otherwise specified (NOS), one of one patient with follicular lymphoma (FL), one of two patients with primary central nervous system lymphoma (PCNSL), and none of the two patients with primary mediastinal B-cell lymphoma (PMBCL) responded.

Double expressor DLBCL tended to show lower ORR (35.3%, 95% CI 16.6–59.4), although 23.5% of double expressor DLBCL patients showed CR. Age, number of lines of previous treatment, International Prognostic Index (IPI) risk at diagnosis, and high baseline lactate dehydrogenase (LDH) levels were not associated with ORR (Fig. 1). Among patients with Bcl-2 positive immunohistochemistry (IHC), those with Myc positive IHC tended to have a lower ORR compared with the patients with Myc negative IHC, but the difference was not statistically significant (35.3% [95% CI 16.6–59.4] vs. 68.4% [95% CI 44.5–85.3], $p = 0.093$). Four patients had double-hit DLBCL, with one patient having all of MYC, BCL2, and BCL6 rearrangements. The one patient with all of the three rearrangements achieved complete remission (CR) with a progression-free survival (PFS) of 32 months. The patient did not experience disease progression until the data cutoff. In the remaining three patients, one patient responded, and the PFS of the three patients were 0.5, 1.4 and 3.3 month.

### Duration of response and survival
The enrolled patients were followed-up for a median duration of 9.1 months from the initiation of the study drugs. The median time to response from the initiation of treatment was 2.0 months (range 1.0–13.4 months). With a median follow-up duration of 19.0 months in responders, the median duration of response was 12.9 months (95% CI 4.9–not available). At the time of data cutoff, 13 patients did not experience progressive disease (Fig. 2A). Nineteen patients with CR, one patient with partial response (PR), and one patient with stable disease received acalabrutinib maintenance. The patient with stable disease received acalabrutinib maintenance despite the protocol for acalabrutinib maintenance being intended for responders because the disease was stable with the R2A regimen after six cycles, and the patient tolerated the regimen well.

The 1-year PFS and overall survival (OS) rate was 33.1% and 67.5%, respectively. The median PFS was 4.4 months (95% CI 3.5–11.6, Fig. 2B). The difference of PFS between nGCB type DLBCL and GCB type DLBCL was not significant (median PFS 4.5 months [95% CI 3.8–15.7] in nGCB type vs. 3.9 months [95% CI 1.4–NA] in GCB type, $p = 0.23$). The median PFS of patients with double-expressor DLBCL was 3.9 months (95% CI, 1.5–NA). Among patients who were Bcl-2 IHC positive, those with Myc positive IHC had a median PFS of 3.9 months (95% CI 1.5–24.0), while those with Myc negative IHC had a median PFS of 5.1 months (95% CI 1.5–24.0); the difference was not statistically significant ($p = 0.29$,

## Table 1 | Patient characteristics

| Characteristic | Number |
|---|---|
| Median age (range) | 67.5 (20 – 87) |
| Sex—no. (%) | |
| Female | 32 (48.5) |
| Male | 34 (51.5) |
| Pathologic diagnosis – no. (%) | |
| DLBCL, nGCB | 47 (71.2) |
| DLBCL, GCB | 11 (16.7) |
| DLBCL, NOS | 3 (4.5) |
| PCNSL | 2 (3.0) |
| PMBCL | 2 (3.0) |
| FL | 1 (1.5) |
| IPI staging at diagnosis—no. (%) | |
| 0–1 | 16 (24.2) |
| 2 | 13 (19.7) |
| 3 | 12 (18.2) |
| 4–5 | 23 (34.8) |
| Not available (PCNSL patients) | 2 (3.0) |
| Previous line of treatment—no. (%) | |
| 1 | 32 (48.5) |
| 2 | 20 (30.3) |
| 3 or more[§] | 14 (21.2) |
| Double expressor—no. (%) | |
| Yes | 17 (25.8) |
| No | 29 (43.9) |
| Not available | 20 (30.3) |
| Double hit—no. (%) | |
| Yes | 4 (6.1)[†] |
| No | 31 (47.0) |
| Not available | 31 (47.0) |
| ECOG performance at the enrollment—no. (%) | |
| 0 | 7 (10.6) |
| 1 | 53 (80.3) |
| 2 or more[†] | 6 (9.1) |
| Baseline LDH—median (IQR) | 333.5 (255.5, 534.8) |

Source data are provided as a Source Data file.

[§]One patient had previously undergone five lines of treatment including one autologous stem cell transplantation. None of the patients had previously received more than four lines of treatment.

[†]One patient had an ECOG performance status score of 3. The other patients had an ECOG performance status score of 2.

[‡]One patient had MYC, BCL2, and BCL6 translocation.

*DLBCL* diffuse large B-cell lymphoma, *ECOG* European Cooperative Oncology Group, *FL* follicular lymphoma, *IPI* International Prognostic Index, *IQR* interquartile range, *GCB* germinal center B-cell type, *LDH* lactate dehydrogenase, *nGCB* non-germinal center B-cell type, *NOS* not otherwise specified, *PCNSL* primary central nervous system lymphoma, *PMBCL* primary mediastinal B-cell lymphoma.

Supplementary Fig. 2). The median OS was not reached (95% CI 23.9–not available, Fig. 2C).

## Adverse events

Thirty-nine patients (59.1%) experienced adverse events (Table 2). Hematological adverse events occurred in 22 patients, with neutropenia being the most frequent, followed by thrombocytopenia. Other adverse events occurred in 27 patients, with skin rashes being the most frequent, followed by pruritus. Grade 3 or higher toxicities were mostly hematologic toxicities, and some were other toxicities. These toxicities were manageable, and no related deaths occurred. Treatment was delayed in 18 patients and dose was reduced in 4 patients due to hematologic toxicities. One patient experienced a drug reaction with eosinophilia and a systemic symptom syndrome. It is not certain whether the symptoms were caused by drugs or disease progression. However, the investigator decided that further administration of the study drugs would not benefit the patient, and the patient was subsequently dropped from of the study.

## Quality of life survey

There were 28 patients with responses available for analyses on the changes from baseline to cycle 2. No significant changes in functional score, global health score, or symptom scale were observed at cycle 2. Responses available for analyses on the changes from baseline to end-of-trial were obtained from 36 patients. No significant changes in the scores were observed in the trial cohort, except for global health score which increased significantly ($p = 0.013$). When the patients were divided into responders and non-responders, patients who responded showed significant increase in global health score ($p = 0.044$), whereas patients who did not respond showed significant decrease in functional score ($p = 0.034$, Supplementary Fig. 3).

## Exploratory biomarker analysis

Samples from 42 patients were available for next generation sequencing (NGS) analysis (Fig. 3A). *MYD88* mutations were noted in seven patients. Among them, six patients with *MYD88* mutations showed an objective response to the R2A regimen, with three CR and three PR patients. The median duration of response was 2.6 months (95% CI 2.1 – NA), but the three patients who experienced CR showed response durations of 15.0, 16.5, and 19.4 months, respectively and did not show disease progression at the data cutoff time point. The median PFS of patients with *MYD88* mutation was 4.5 months (95% CI, 3.5 – NA). The three patients who experienced CR showed PFS of 18.2, 19.0 and 21.1 months, respectively, at the data cut-off time point. Two patients harboring *PIM1* mutation did not respond to the treatment. As *CD79A*, *CD79B*, and *CARD11* mutations have been implicated in potential resistance to ibrutinib in the previous literature[15], we investigated these mutations. Out of two patients with *CD79B*, one did not respond to the treatment, and the other showed a DoR of only 2.1 months even though the patient harbored *MYD88* mutation. The two patients harboring *CD79A* mutation did not respond to treatment, whereas the patient harboring *CARD11* mutation showed CR.

A total of two patients were classified into MCD subtype by LymphGen classification[16], and the remaining 40 patients were unclassifiable. Total of nine patients were classified into specific subtype by LymphPlex classification (Four EZB-like-MYC-, three TP53[Mut], one BN2-like and one MCD-like subtype)[17]. The outcomes of patients according to these subtypes are summarized in Table 3. Notably, three patients classified into subtypes that are known to have NF-κB activation (MCD, MCD-like, and BN2-like subtype) responded to the R2A regimen. There was no difference in survival outcomes according to each subtype classification (Supplementary Fig. 4).

A total of 30 patients had available tissue for RNA sequencing, and expression profiling for cell-of-origin subtyping was conducted in these patients. Among them, 13, 11, and 6 patients were classified into the activated B-cell (ABC), GCB, and unclassifiable subtypes, respectively. The ORR was 69.2% (95% CI 41.3–88.8), 54.5% (95% CI 26.0–80.1), and 83.3% (95% CI 41.0–99.2) in each of the ABC, GCB, and unclassifiable subtype, respectively, with no significant differences ($p = 0.613$). There were no significant differences in PFS and OS according to the cell-of-origin subtype (Supplementary Fig. 5).

BTK expression was evaluated in the tissue samples from 11 patients using single-molecule fluorescence imaging. Six patients showed high BTK expression, whereas the remaining five showed low expression. Remarkably, three of the four patients who responded to treatment (two with complete response and one with partial response) had high BTK expression, while one of the five patients with low BTK

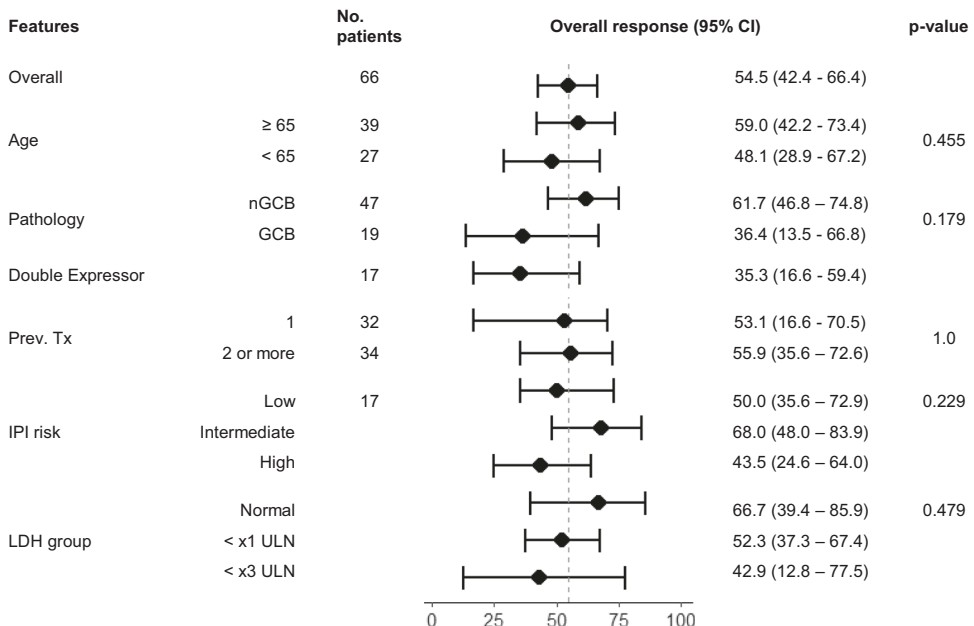

**Fig. 1 | Forrest plot of response rate according to demographic features.** The center black dot in each row represents the response rate, and the line with vertical whiskers show 95% confidence interval. The 95% confidence intervals were calculated using the Blythe-Still-Casella method. A two-sided Fisher's exact test was used to compare the response rates between the groups, without adjusting for multiple comparisons. Source data are provided as a Source Data file. CI confidence interval, GCB germinal center B-cell type, IPI International Prognostic Index, LDH lactate dehydrogenase, nGCB non-germinal center B-cell type, Prev. Tx number of previous treatment lines, ULN upper limit of normal.

expression responded. Additionally, of the six patients with high BTK expression detected by this method, five were also identified as BTK-high by D3H5 IHC analysis, and all six were identified as BTK-high by D6T2C IHC analysis. Among the five patients with concordant high BTK expression status, three responded to BTK inhibitors. Patients with high BTK expression on single-molecule fluorescence imaging tended to have a longer PFS (median PFS 5.2 months [95% CI, 4.5–NA]) than those with low BTK expression, but the difference did not reach statistical significance (median PFS 2.0 months [95% CI, 0.8–NA]; $p = 0.055$; Supplementary Fig. 6).

## Discussion

In this phase II clinical trial on patients with R/R B-cell NHL, mostly DLBCL, a combination of acalabrutinib with rituximab and lenalidomide was effective, with manageable toxicities and a fair quality of life. The ORR of the trial regimen (54.5%) was higher than that of BTK inhibitor acalabrutinib monotherapy (24%) for DLBCL reported by Strati et al.[18]. and that of ibrutinib therapy (35%) reported by Graf et al.[19], with. In addition, the ORR was also higher than that in the previous literature on the rituximab/lenalidomide combination; 33% in Wang et al.[20], and 35% in Zinzani et al.[21]. Although direct comparisons between this trial and other trials should not be interpreted as our results being superior, we highlight the potential synergistic effect of the study drugs in patients, consistent with the results of in vitro studies[9].

Although no clinical factors associated significantly with the response to the R2A regimen, nGCB-type DLBCL tended to show a higher response rate than GCB-type DLBCL. This was expected, as ABC type DLBCLs reportedly shows frequent chronic active B-cell receptor signaling and somatic mutations affecting immunoreceptor tyrosine-based activation motif signaling[21], and BTK inhibition can effectively block the B-cell receptor signaling pathway[22]. In clinical trials of BTK monotherapy, the ORR of GCB-type DLBCL has been consistently lower (approximately 5–20%) than that of ABC type DLBCL[8,18], and this tendency seems consistent in the R2A regimen. However, two patients with GCB-type DLBCL experienced CR. As the combination of

rituximab and lenalidomide may enhance tumor susceptibility to acalabrutinib, further preclinical studies to find subset of GCB-type DLBCL that may be dependent on the BTK signaling may be needed.

There are concerns about the toxicity of combination regimens. Both acalabrutinib and lenalidomide are associated with hematologic toxicities; therefore, we included a safety cohort of the first three patients treated with reduced lenalidomide. In the entire study cohort, one-third of the patients experienced hematologic toxicities, which is consistent with previous clinical trials. The reported incidences of the hematologic toxicities in two previous clinical trials on ibrutinib, rituximab, and lenalidomide combination for R/R DLBCL reported incidence ranging from 20 to 44%[11,23]. Another clinical trial on R2A for mantle cell lymphoma reported hematologic toxicity in up to 38% of patients[24]. Notably, in our study, the toxicities were manageable and most of the patients recovered within a cycle. In addition, the quality of life of these patients was generally acceptable, although responses to treatment seemed to have impact and there may be a selection bias towards patients who responded to the survey.

Recently, DLBCL treatment has begun to include CAR T-cell therapy. An anti-CD19 CAR T-cell therapy agent, axicabtagene ciloleucel, showed ORR of 83% and PFS of 24.9 months in a randomized clinical trial[25]. Another anti-CD19 CAR T-cell therapy agent, lisocabtagene maraleucel, showed ORR of 73%[26]. However, the application of CAR T-cell therapies may be hindered by high costs and reimbursement issues[7]. In Republic of Korea, tisagenlecleucel was only recently approved for relapsed/refractory DLBCL patients in 2022, yet ongoing financial and administrative challenges persist[27]. Furthermore, the promising results of CAR T-cell therapies come at the expense of severe toxicities, including cytokine storms and encephalopathy, which pose potential fatal risks and demand intensive management[25,26]. Therefore, in situations where CAR T-cell therapies are not available, the R2A regimen may be considered for relapsed/refractory DLBCL patients. Recently, several BiTEs have shown promising efficacies. Based on the results of R2A, a combination of a BTK inhibitor and a CD20-targeting BiTE may be effective in overcoming the potential limitation of CAR T-cell therapy and failures to CD19-

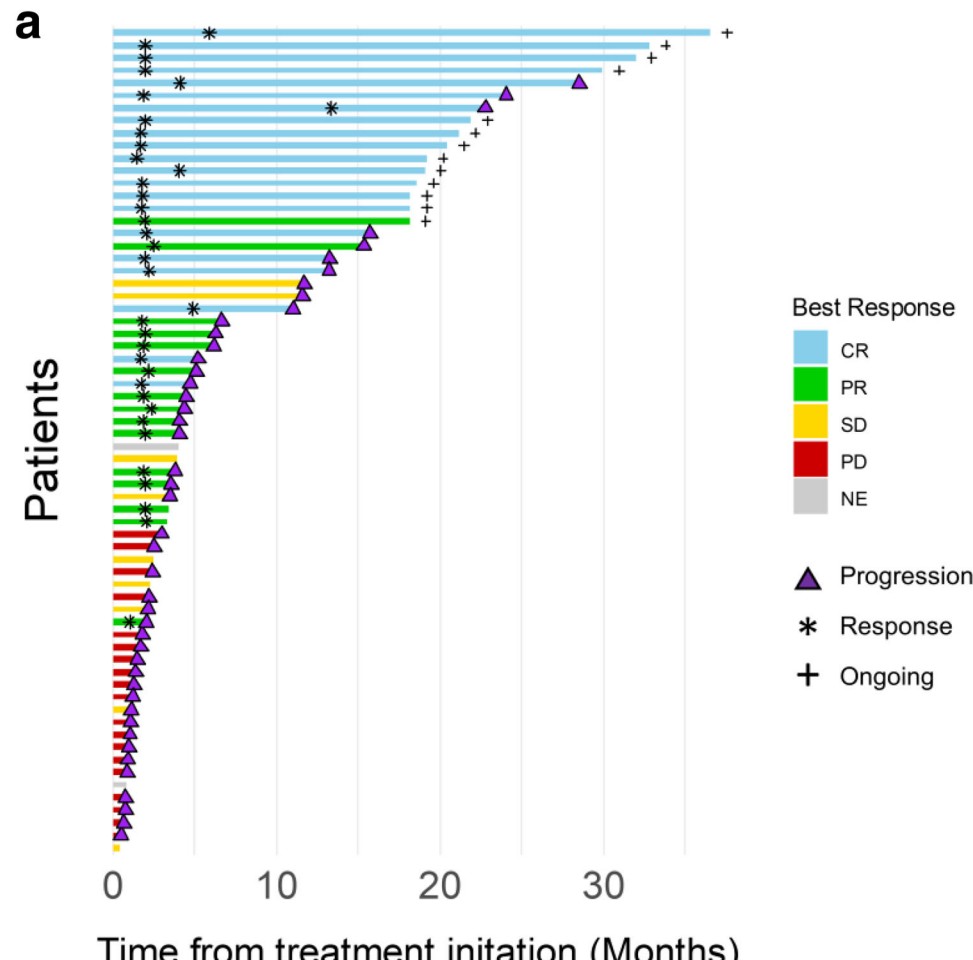

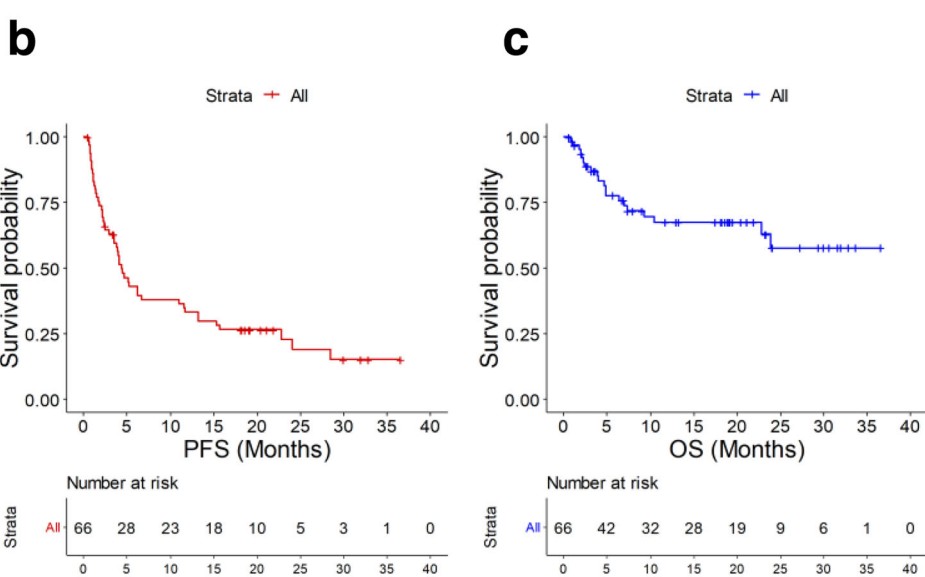

**Fig. 2 | Duration of response and survival. a** Swimmer plot of patients enrolled in the trial. Each row represents a patient and the length of each bar represents the time from treatment initiation. **b** Kaplan-Meier survival curve for progression free survival. **c** Kaplan-Meier survival curve for overall survival. The cross marks on the Kaplan-Meier curves depict the censored data. Source data are provided as a Source Data file. CR complete response, OS overall survival, PFS progression-free survival, PR partial response, SD stable disease, PD progressive disease, NE not evaluable.

**Table 2 | Adverse events**

| Adverse event term | Any grade | Grade 1 | Grade 2 | Grade 3 | Grade 4 | Grade 5 |
|---|---|---|---|---|---|---|
| Hematologic toxicity | | | | | | |
| Neutrophil count decreased | 21 | 1 | 0 | 8 | 12 | 0 |
| Platelet count decreased | 6 | 1 | 2 | 1 | 2 | 0 |
| Febrile neutropenia | 2 | 0 | 0 | 2 | 0 | 0 |
| Anemia | 1 | 0 | 0 | 1 | 0 | 0 |
| Eosinophilia | 1 | 0 | 1 | 0 | 0 | 0 |
| Other toxicity | | | | | | |
| Skin rash | 17 | 2 | 10 | 5 | 0 | 0 |
| Pruritus | 6 | 1 | 4 | 1 | 0 | 0 |
| Urticaria | 2 | 0 | 0 | 2 | 0 | 0 |
| Abdominal pain | 2 | 2 | 0 | 0 | 0 | 0 |
| Diarrhea | 2 | 0 | 2 | 0 | 0 | 0 |
| Neuropathy | 2 | 0 | 2 | 0 | 0 | 0 |
| Anorexia | 1 | 0 | 1 | 0 | 0 | 0 |
| Fatigue | 1 | 0 | 1 | 0 | 0 | 0 |
| Fever | 1 | 1 | 0 | 0 | 0 | 0 |
| Myalgia | 1 | 0 | 1 | 0 | 0 | 0 |
| Headache | 1 | 0 | 1 | 0 | 0 | 0 |
| Edema (Face) | 1 | 0 | 1 | 0 | 0 | 0 |
| Oral mucositis | 1 | 0 | 1 | 0 | 0 | 0 |
| Nausea | 1 | 0 | 0 | 1 | 0 | 0 |
| Vomiting | 1 | 0 | 0 | 1 | 0 | 0 |
| Constipation | 1 | 0 | 1 | 0 | 0 | 0 |
| Hypertension | 1 | 0 | 1 | 0 | 0 | 0 |
| Invasive pulmonary aspergillosis | 1 | 0 | 0 | 1 | 0 | 0 |
| Lung infection | 1 | 0 | 0 | 1 | 0 | 0 |
| Cramp on leg | 1 | 0 | 1 | 0 | 0 | 0 |
| Zoster | 1 | 0 | 1 | 0 | 0 | 0 |
| Alanine aminotransferase elevation | 1 | 0 | 0 | 1 | 0 | 0 |
| Infusion related reaction | 1 | 0 | 1 | 0 | 0 | 0 |

targeting CAR T-cell therapy[28]. We are currently conducting a clinical trial to evaluate the combination of glofitamab, poseltinib, and lenalidomide for patients with R/R DLBCL (ClinicalTrials.gov identifier: NCT05335018).

We sought to identify potential biomarkers associated with the response to the R2A regimen. Patients with GCB-type DLBCL and/or double expression showed a lower response. However, because some of these patients showed good responses, including CR, excluding them from the R2A regimen would not have been appropriate, and a more precise biomarker appeared necessary. Meanwhile, we performed BTK IHC on a small set of patients with tissues available for IHC. Although the numbers were too small to show any significance, we observed that patients who responded showed strong BTK IHC staining. Therefore, BTK IHC is warranted in more patients to draw conclusions on the association of BTK IHC with BTK inhibition.

We also observed that patients harboring *MYD88* mutations elicited fair responses to the R2A regimen. This is explained by a previous study that reported phosphorylated BTK in *MYD88* mutated plasma cells, suggesting that BTK is a downstream target of mutated *MYD88*

signaling[29]. The importance of *MYD88* mutation in BTK is also implicated in the genetic study of DLBCL, which showed enrichment of the mutation in activated B-cell type DLBCL[13]. Further studies should determine whether the efficacies observed in patients with *MYD88* mutation are driven by acalabrutinib only or have been augmented by the combination regimen.

While no sufficiently recurrent mutations were noted in patients with a poor response to the R2A regimen, we observed some hypothetical mutations possibly associated with resistance to the R2A regimen. For example, a gain-of-function mutation in *PIM1*, which encodes a kinase associated with the tumorigenesis of hematologic malignancies and is associated with resistance to BTK inhibitors, was observed in two patients resistant to the R2A regimen[30]. The three of four patients who harbored *CD79A* or *CD79B* were resistant to the R2A regimen, whereas the remaining patient harboring *MYD88* mutation showed a shorter response. *CD79A/CD79B* mutations may impart resistance to the BTK inhibitor response[15]. In contrast, patient harboring *CARD11* mutation, which is downstream signal of BTK, did not show resistance to the R2A regimen, which may implicate that rituximab/lenalidomide combination allowed overcoming BTK inhibitor resistance[31].

Due to the limitation of the targeted sequencing, most of the samples could not be classified into specific subtypes as proposed by previous literature. However, we observed that subtypes with NF-κB activation, which is related to B cell receptor signaling and BTK pathway, responded to the R2A regimen[16,17]. Since only a few samples were evaluated in this study, conducting further research using whole-exome sequencing to assess the association of the LymphGen classification with the R2A regimen would be valuable. This is particularly important as the combination regimen is likely to be effective in specific subgroups of lymphoma with an enrichment in the BTK pathway.

This study had some limitations. First, this was a single-arm phase II clinical trial; therefore, there was no explicit comparison. The ORR comparisons with previous studies warrant further randomized controlled clinical trials comparing R2A regimens with other regimens to determine the differences between their efficacies. Secondly, the OS data were not sufficiently mature to assess the prognosis of these patients. Third, most patients included in this study had DLBCL, and only one of four patients with PMBCL or FL responded to the R2A regimen. Application of the R2A regimen to R/R B-cell NHL, other than DLBCL, requires caution. In addition, further studies on the R2A regimen for DLBCL would require the stratification of patients according to GCB or nGCB subtypes, as the ORR differences between the two types may have a significant impact. Fourth, all patients in this study did not receive CAR T-cell therapy prior to R2A. Therefore, the efficacy of R2A in patients who received prior CAR T-cell therapy cannot be evaluated in this study.

In conclusion, the combination of acalabrutinib, rituximab, and lenalidomide is an acceptable regimen for R/R B-cell NHL with tolerable toxicities. Further studies are warranted to evaluate the synergistic efficacy of similar combinations, such as combination of a BTK inhibitor and a BiTE. In addition, biomarkers based on BTK IHC and genetic classification should be developed to identify patients likely to respond to the R2A regimen.

## Methods
### Patients
Patients diagnosed with DLBCL, including GCB type and nGCB type, PMBCL, transformed FL, and small lymphocytic lymphoma with Richter transformation were enrolled. Patients should also have experienced relapse or refractory to at least one line of treatment if they were ineligible for autologous stem cell transplantation and at least two lines of treatment if they were candidates for autologous stem cell transplantation. Previous treatments should have included anti-CD20 based chemotherapy. Patients with central nervous system involvement could also be enrolled. The sex was not considered in study design and analyses as there is no definite known sex-difference

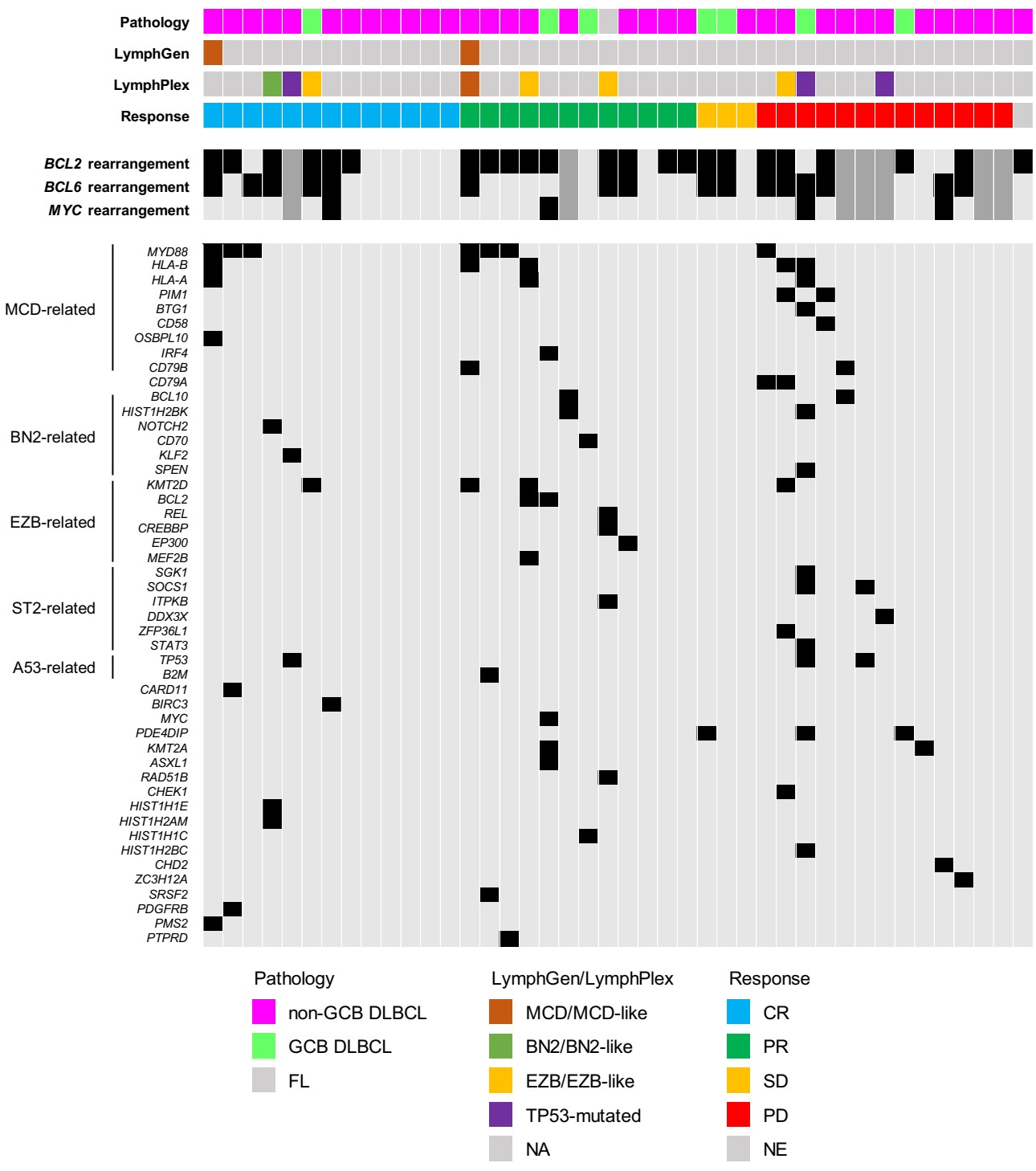

**Fig. 3 | Results of targeted gene next-generation sequencing, classification, and outcome according to mutation status.** Landscape plot of next generation sequencing results. The genes related to each LymphGen subtype are annotated in the left. There were seven patients whose data on the BCL2, BCL6, and MYC rearrangements were not available, which are marked with dark gray color. Source data are provided as a Source Data file. CR complete remission, DLBCL diffuse large B-cell lymphoma, FL follicular lymphoma, GCB germinal center B-cell type, NE not evaluable, PD progressive disease, PFS progression free survival, PR partial response, SD stable disease.

in patients with relapsed/refractory B-cell lymphoma. Further detailed eligible and ineligible criteria are presented in Supplementary Note 1.

**Trial design and treatment**

This was a prospective, open-label, single-arm, phase II study that assessed the efficacy and safety of a combination regimen of acalabrutinib with rituximab and lenalidomide. A treatment cycle consisted of days 1–28, during which rituximab 375 mg/m2 was administered intravenously on day 1, lenalidomide 20 mg orally once daily from days 1 to 21, and acalabrutinib 100 mg orally twice daily from days 1 to 28. The cycle was repeated every four weeks for up to six cycles if the disease did not progress during this period. Those who responded to the R2A regimen, defined as PR or CR, and completed the six cycles of the combination were offered to receive maintenance treatment with

**Table 3 | Efficacies of patients who were classified into specific subtypes**

| Patient | LymphGen classification | LymphPlex classification | Best objective response | PFS (months) | OS (months) |
|---|---|---|---|---|---|
| PUH-002 | MCD | Unclassifiable | CR | 18.2§ | 18.2† |
| BPH-003 | MCD | MCD-like | PR | 4.1 | 7.4 |
| SUH-001 | Unclassifiable | BN2-like | CR | 36.6§ | 36.6† |
| BPH-001 | Unclassifiable | EZB-like | PR | 6.3 | 31.2† |
| KDH-002 | Unclassifiable | EZB-like | CR | 13.3 | 23.9 |
| KGH-004 | Unclassifiable | EZB-like | PR | 4.4 | 13.3† |
| SUH-008 | Unclassifiable | EZB-like | PD | 2.5 | 4.9 |
| JBH-002 | Unclassifiable | TP53$^{Mut}$ | CR | 19.2§ | 19.2† |
| KDH-003 | Unclassifiable | TP53$^{Mut}$ | PD | 1.4 | 2.3 |
| SUH-014 | Unclassifiable | TP53$^{Mut}$ | PD | 0.6 | 27.2† |

§These patients had not experienced progressive disease until the data cutoff time.
†These patients were alive at the time of the data cutoff.
*CR* complete remission, *PFS* progression-free survival, *PR* partial response, *OS* overall survival

acalabrutinib monotherapy up to 1 year. To address safety concerns, we enrolled the first three patients in the safety cohort and started with a reduced dose of lenalidomide (15 mg) once daily in the first cycle. Details of the trial protocols are available in Supplementary Note 1.

**Assessments**
Disease assessments were done with computed tomography (CT) scan with contrast of the chest, abdomen, pelvis, and other disease sites along with positron emission tomography/CT (PET)/CT scans. Response assessments were evaluated based on the Lugano criteria[32]. Patients were contacted approximately every 3 months to assess disease status and survival.

Patient characteristics were recorded at the time of enrollment. Pathologic reports by board-certified pathologist in each center were collected. The reports consisted of pathologic diagnosis, IHC reports and fluorescence in situ hybridization reports, which include B-cell lymphoma 2 (Bcl-2), B-cell lymphoma 6 (Bcl-6), Myc, and Ki-67 protein expressions, translocations of Bcl-2, Bcl-6, and Myc. The assays in each patient were performed as per the clinical decisions from each center.

Adverse events were reported by each investigator. The terms and severity of the adverse events were recorded according to the Common Terminology Criteria for Adverse Events version 5.0[33]. quality of life of each subject was assessed using the EORTC-QLQ-C30[34]. The EORTC-QLQ-C30 was administered at the time of enrollment, after 2nd cycle, and at the time of treatment termination.

Details on the assessments are available in Supplementary Note 1.

**Outcomes**
The primary outcome of this study was objective response rate defined as the proportion of patients whose best objective response according to the Lugano criteria was PR or CR. Secondary outcomes included CR rate, DoR, PFS, OS, and treatment-related adverse events.

**Post-hoc exploratory biomarker analysis**
ORR and PFS were evaluated according to the tumor IHC status. Myc expression ≥70% in tumor cells suggested Myc amplification[35]. IHC staining for Btk was performed in whole sections of each representative FFPE block in all cases. We measured BTK expression levels using single-molecule fluorescence imaging, calibrated the results to the area of the formalin-fixed, paraffin-embedded specimens, and used a threshold of 10000 to distinguish between high and low BTK levels. We also performed conventional Btk IHC using BTK antibodies (D3H5 and D6T2C, Cell Signaling Technology), and Btk was scored on five levels according to intensity (0, completely negative; 1, faint; 2, weak; 3, intermediate; 4, strong) by an experienced hematopathologist (J.C.). Scores of 2 or higher were classified as BTK high and BTK low otherwise.

For targeted tumor NGS analysis, pre-treatment tumor tissues were obtained from the enrolled patients. The list of genes for targeted DNA sequencing and RNA sequencing are available at Supplementary Tables 1, 2. For classification of tumors according to genetic status, we used algorithms proposed by LymphGen classification[16] and LymphPlex[17], which allow classification similar to LymphGen classification with simplified 38-gene algorithm.

Cell-of-origin subtyping using RNA sequencing was conducted on tumor samples for which whole transcriptome sequencing data were available. We followed the procedures described in previous literature[36].

**BTK immunohistochemistry expression measurement**
The list of antibodies used for BTK immunohistochemistry in this study is available at Supplementary Table 3. We used the cell extract from TMD8 cell as positive control. To prepare FFPE slides for analysis, we followed a rigorous protocol that involved multiple steps. Firstly, we deparaffinized the slides using xylene and sequentially rehydrated them with 100%, 95%, and 70% ethanol, followed by 18MOhm water. Next, we collected the rehydrated specimens in AFA tubes and resuspended them in an extraction buffer consisting of 1% SDS and 50 mM Tris-HCl (pH=8.0). The specimens were homogenized for 6 min using Covaris M220, followed by a 1-hour incubation of the AFA tubes at 99 °C and 750 rpm in an Eppendorf thermomixer for antigen retrieval and protein extraction. We then transferred the resulting extracts to Eppendorf tubes and centrifuged them at $15000 \times g$ to collect the supernatants, which we diluted 10-fold with a dilution buffer consisting of 1% TX100, 50 mM Tris-HCl (pH=8.0), and 150 mM NaCl to reduce the SDS concentration.

**RNA& DNA extraction from tissues**
Total RNA from tissue was extracted using RNeasy mini kit from Qiagen (Qiagen, USA), according to the manufacturer's recommendations. gDNA from tissue was extracted using QIAamp DNA tissue kit (Qiagen, USA), according to the manufacturer's recommendations.

**Total RNA Sequencing**
We used 100 ng total RNA from all subjects to prepare sequencing libraries with by using the TruSeq stranded total RNA sample preparation kit (Illumina, CA, USA) which combines RiboZero rRNA depletion with a stranded specific method similar to the dUDP method. Quality of these cDNA libraries was evaluated with the Agilent 2100 BioAnalyzer (Agilent, CA, USA). They were quantified with the KAPA library quantification kit (Kapa Biosystems, MA, USA) according to the manufacturer's library quantification protocol. Following cluster amplification of denatured templates, sequencing was progressed as paired-end (2 × 150 bp) using Illumina NovaSeq6000 platform.

## Targeted DNA sequencing

500 ng of genomic DNA from each sample were sheared, and the size of 250–300 bp fragmented DNA were purified by Beckman Ampure Bead. Libraries were constructed using ACCEL-NGS 2 S DNA library kit from Swift Biosciences. Library preparation starts with Repair I, II and Ligation I, II. The indexed adapter is added during the Ligation step I to allow pooling of multiple libraries for hybridization and sequencing. PCR amplification was performed for 4 cycles and the concentration of the pre-libraries were quantified by Qubit double strand BR assay. IDT targeted probe (258 genes) captures were hybridized as pools of 8 samples using 150 ng of library input, respectively. All hybridization and post captured washes were performed according to IDT protocol. Post PCR was performed using the Kapa HiFi DNA polymerase for 11-12 cycles. After the libraries were constructed, the Agilent TapeStation 4200 can be used to check the fragment size and concentration with the D1000 screen tape. A size distribution is between 380-430 bp and the final concentration is over than 5 ng/ul. Moreover, each pooling samples were quantified using KAPA Library Quantification Kit. Sequencing was performed on the Illumina NovaSeq6000 platform with 2×150 read lengths by following the manufacturer's protocols.

## Targeted RNA sequencing

We used 100 ng total RNA from all subjects to prepare sequencing libraries with by using the Swift RNA sample preparation kit (Swift-bioscience, USA). The pre-libraries were quantified by Qubit double strand BR assay. IDT targeted probe (86 genes) captures were hybridized as pools of 8 samples using 150 ng of library input, respectively. All hybridization and post captured washes were performed according to IDT protocol. Quality of these cDNA libraries was evaluated with the Agilent 2100 BioAnalyzer (Agilent, CA, USA). They were quantified with the APA library quantification kit (Kapa Biosystems, MA, USA) according to the manufacturer's library quantification protocol. Following cluster amplification of denatured templates, sequencing was progressed as paired-end (2 × 150 bp) using Illumina NovaSeq6000 platfrom.

## Data preprocessing, quality control analysis, and control cohort

Targeted sequencing reads for the lymphoma samples were demultiplexed using Illumina's bcl2fastq (v2.17.1.14) to generate FASTQ files. We used SeqPrep for adapter trimming (default settings) and Sickle (v1.33) for low BQV base trimming (minimum average BQV = 20). Subsequently, trimmed FASTQ files were submitted to GATK best practice pipeline, which includes alignment to the hg19 reference with BWA-MEM (v0.7.10). For all samples, duplicate marking and sorting was done using PICARD (v1.94) MarkDuplicates, followed by indel realignment and base quality score recalibration using GATK Light (v2.3.9) and duplicate marking again, resulting in a final coordinate sorted BAM per sample as an analysis ready BAM. Duplication metrics and BAM quality metrics were computed using PICARD (v1.94; MarkDuplicates, CalculateHsMetrics, CollectGcBiasMetrics). Analysis ready BAM files for the analyzed cohort were qualified with depth of coverage (average DOC 1 555x; 835x ~ 2843x), which guarantees 2% VAF limit of Detection.

## Somatic mutation calling and filtering

Analysis ready BAM files were processed through somatic variant calling pipeline that consists VarDict[37], Mutect2 (4.1.4.1)[38], SNVer (0.4.1)[39], for calling SNVs, insertion, and deletion. To achieve comprehensive somatic variant calling, we enforced union between variant callers. Technically, the requirements to be positive SNVs/Indels were total reads ≥20, Alt reads ≥20 (positive & negative ≥ 10, respectively), and VAF ≥ 2% and 30%. Even though VAF > 30%, the variants with COSMIC hematological criteria and median batch VAF < 45% were selected. We further filtered common germline variants both with MAF > 0.2% in gnomAD and without COSMIC evidence. Finally, final variants were curated by IGV review to filter out potential artifacts driven by PCR, highly homologous regions, and repeat regions.

## Trial oversight

This investigator-initiated trial was designed by the authors. The trial protocol and amendments were approved by the appropriate institutional review board of each participating center, including the approval by institutional review board of Seoul National University Hospital where the primary investigator is affiliated (IRB no. H-1911-092-988). Informed consent was obtained from all participants before enrollment, and all participants were given the opportunity to withdraw their consent at any time during the trial. The trial was posted publicly at the Ministry of Food and Drug Safety of the Republic of Korea on March 4th, 2019, which was prior to the enrollment of the first patient on July 9th, 2019. The trial was also posted at ClinicalTrial.gov (identifier: NCT04094142) on September 18th, 2019, which had been delayed because the investigators missed a request for additional information and subsequently submitted the required details to the registration system later.

The trial was conducted in accordance with the protocols and standards of Good Clinical Practice. All authors have access to the data related to this manuscript. The manuscript was available to the drug-supplying companies AstraZeneca, Celltrion, and Samyang, but they had no authority to change the manuscript. Other than the investigational medicinal products, no additional compensation was provided to the participants.

## Statistics & reproducibility

For the sample size calculation, we assumed that the ORR for our study would reach approximately 45%, considering the efficacy of combination regimens with rituximab and lenalidomide and monotherapy with ibrutinib for salvage chemotherapy in patients with R/R DLBCL. Under these assumptions, the number of patients required to draw conclusions with an alpha error rate of 5% and power of 80% was 60. Considering a dropout rate of 10%, 66 patients were required.

The 95% CI for the ORR and CR rates were calculated using the Blythe-Still-Casella method. Fisher's exact test was used to compare the ORR between the groups. The median DoR, PFS, and OS were calculated using the Kaplan-Meier method. Comparisons between repeatedly measured values were performed using the Wilcoxon signed-rank test. The data cutoff time for the study was May 15th, 2022, which was decided by primary investigator after the completion of R2A administration. Survival rates were compared using the log-rank test. Statistical analyses were performed using R software version 4.1.0.

## Reporting summary

Further information on research design is available in the Nature Portfolio Reporting Summary linked to this article.

# Data availability

Data are available within the article, Source Data, and Supplementary Information. All non-commercial requests to the corresponding author for other raw and analyzed data and materials, which include specific age, treatment date, follow-up date and death date, will be reviewed by the corresponding author at any time after the publication. Such patient-related data not included in the paper is subject to patient confidentiality and will be shared only after discussion with the institutional review board, which may take months. The raw data of sequencing data in this study is available at BioProject database (BioProject ID: PRJNA1024646). Source data are provided with this paper.

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

## Acknowledgements

This research was supported by an externally sponsored research program (ESR-18-13701) by AstraZeneca (provision of acalabrutinib), Celltrion Healthcare Co., Ltd (provision of rituximab biosimilar), and Samyang Holdings (provision of lenalidomide). The funders have no role in the design, performance, data collection, analysis, drafting the manuscript, and decision to submit and publish the manuscript of this trial. The trial process was supported by Consortium for Improving Survival of Lymphoma (CISL) in Republic of Korea. Chansub Lee from GenomeOpinion Inc. assisted on the cell-of-origin analysis.

## Author contributions

C.P. designed research, performed research, collected data, analyzed and interpreted data, performed statistical analysis, and wrote the manuscript. H.S.L., K.-W.K., W.-S.L., Y.R.D., J.-Y.K., H.-J.S., S.-Y.K., J.H.Y., S.-N.L., J.-O.L., D.-H.Y., J.M.B. and S.-S.Y. performed research, collected data, and reviewed the manuscript. H.J., B.C., J.L., and C.H.S. performed research, collected data, analyzed and interpreted data, performed

statistical analysis, and wrote the manuscript. Y.K. designed research, performed research, collected data, analyzed and interpreted data, and wrote the manuscript.

## Competing interests

C.H.S. and Y.K. are founder and stockholder of GenomeOpinion Incorporation. J.L. is employee of GenomeOpinion Incorporation. H.J. and B.C. are employee of PROTEINA corporation. The other authors have no conflict of interest.

## Additional information

[1]Department of Internal Medicine, Seoul National University Hospital, Seoul, Republic of Korea. [2]Department of Internal Medicine, Kosin University College of Medicine, Gospel Hospital, Pusan, Republic of Korea. [3]Department of Internal Medicine, Korea University College of Medicine, Anam Hospital, Seoul, Republic of Korea. [4]Department of Internal Medicine, Busan Paik Hospital, Pusan, Republic of Korea. [5]Department of Internal Medicine, Keimyung University Dongsan Medical Center, Daegu, Republic of Korea. [6]Department of Internal Medicine, Jeonbuk National University Medical School, Jeonju, Republic of Korea. [7]Department of Internal Medicine, School of Medicine, Pusan National University, Pusan, Republic of Korea. [8]Department of Internal Medicine, Department of Hematology/Oncology, KonKuk University Hospital, KonKuk University, Seoul, Republic of Korea. [9]Department of Internal Medicine, Division of Hematology-Oncology, Department of Medicine, Chung-Ang University, Seoul, Republic of Korea. [10]Department of Internal Medicine, Department of Hematology-Oncology, Inje University Haeundae Paik Hospital, Pusan, Republic of Korea. [11]Department of Internal Medicine, Seoul National University Bundang Hospital, Seongnam, Republic of Korea. [12]Department of Internal Medicine, Chonnam National University Hwasun Hospital, Hwasun, Republic of Korea. [13]Department of Biomarker Discovery, PROTEINA Co., Ltd, Seoul, Republic of Korea. [14]GenomeOpinion Inc., Seoul, Republic of Korea. ✉e-mail: go01@snu.ac.kr

