## [Peer Review File · Nature Communications]

Combination of acalabrutinib with lenalidomide and rituximab in relapsed/refractory aggressive B-cell non-Hodgkin lymphoma: a single-arm phase II trialREVIEWER COMMENTS

Reviewer #1 (Remarks to the Author):

Park et al. present data of a phase 2 study of the BTK inhibitor acalabrutinib in combination with rituximab and lenalidomide in patients with relapsed/refractory diffuse large B-cell lymphoma (DLBCL). The authors report promising efficacy and overall a manageable toxicity profile. Furthermore, the authors report response rates in molecularly defined subgroups of DLBCL.

Major criticism:

1. How many patients received prior CAR T-cell therapies?
2. How many patients had lymphomas with double or triple hit constellation?
3. In their NGS analyses which genes were investigated?
4. How was the response in unclassifiable DLBCL according to the Lymphgene subtypes?
5. The authors should perform gene expression profiling to determine ABC and GCB DLBCL and correlate these findings to response and PFS, OS, etc.
6. How was the assay determining BTK expression established. Which controls were used?

Reviewer #2 (Remarks to the Author):

- multiple of the non-significant results were over-interpreted. Interpreting as a wide confidence interval and large p-value as indicated that a group "tended to" do anything feels like it doesn't recognize the limited precision available for many of the results presented. Presenting the results and confidence intervals without additional narrative description would be appropriate.

- given the very small N in biomarker subgroups the conclusion in the abstract biomarker enrichment designs are warranted seems like an over-statement.

Reviewer #3 (Remarks to the Author):

Dr. Koh and colleagues have conducted an important clinical trial evaluating the combination of rituximab, lenalidomide, and acalabrutinib. The rationale of the combination is strong, and the trial was well designed.

Introduction:

1. The first sentence is confusing as it does not define where CR rates are 76% - with first line therapy?
2. The "significant portion" with refractory disease is vague, please define as significant could be interpreted broadly.
3. Language - it is difficult to define "an ideal regimen" for R/R B-cell NHL broadly as R2A. This language should be toned down.
4. The references cited here are unusual. Citing one paper about a prospective CAR T_cell trial but not others, then citing a review papers about commercial CAR T-cell vs other therapies but not a review paper about DLBCL in general. Suggest closer attention to which papers are cited here.

Materials and Methods

1. The authors mention patients "failed at least one line of treatment". This language is not acceptable - the patient did not fail the treatment, but instead the treatment failed the patient. We should not blame our patients for treatment failure. Change to some version of disease relapsed or refractory to at least 1 line of treatment.
2. Presumably second line CAR T-cell therapy was not available as it is not mentioned in the exclusion for patients who had only 1 prior therapy (but ASCT is). Correct?
3. Why are there no details about the treatment doses and schedule in the manuscript, but there are details about a safety cohort?
4. The definition of "pathology" is not sufficient. Were assays to determine cell of origin or RNA sequencing conducted? If so, which assays?
5. Was CNS involvement excluded?

Results

1. BCL6 expression is mentioned prior to Double expressor - please confirm the authors are not including BCL6 expressing via IHC as a part of "double expressor", as this term only applies to MYC and BCL2 protein expression.
2. Were any patients double hit with MYC and BC2 or BC6 translocations?
3. "The enrolled patients received lenalidomide 20mg" - this needs to be clarified with an English editor - I presume this means that all subsequent patients received 20mg from cycle 1.
4. One patient with PR received acalabrutinib maintenance, why not the other patients with PR? Why did 2 patients with CR not receive the acalabrutinib maintenance?
5. It is difficult to understand how only 33% of patients had any hematologic toxicity. This rate seems very low - were the other 66% of patients completely free from cytopenias?
6. It is difficult to understand how none of the patients had any change in quality of life on the three metrics mentioned. Even patients with lymphoma that was refractory to treatment had no change to quality of life? Even the patient that came off due to the eosinophilia? This raises questions of how many patients had pre and post therapy assessment of QoL.
7. The sentence "all EZB-related gene-mutated and ST2-related gene wild-type patients (N = 5) 293 responded to R2A, whereas none of the wild-type and ST2-related gene-mutated patients" is worded incorrectly. What is the second use of wild type to mean - EZB-related wild type? I have not seen these lymphGen subsets described as mutated or wild type in other manuscripts. It may be more clear what the authors are implying to say genes associated with EZB or ST2, and not use the label of EZB-wildtype or ST2-mutated.
8. "LymphGene" is mis-spelled
9. Comparing mutations between the subsets as the authors are doing (WT here, mutation there) is not standard. Typically patients are assigned to one subtype, and not classified based upon wild type status in other subtypes. This is concerning for multiple comparisons fallacy where random combinations of subsets could appear significant. This analysis is deeply flawed, and should not be presented as though the "Subset" of EZB-WT/ST2-WT is favorable. Which subtype are those patients? It is acceptable to show that EZB subtype or ST2 subtype have outcomes, but to lump together all other subtypes and call them effectively "not EZB or ST2" is not appropriate.

Discussion

1. The authors compare their results with other studies in other subtypes of B NHL, when this trial was nearly all DLBCL. These comparisons are flawed due to different types of NHL having differential responses, but the authors present them as comparable.
2. The sentence "Although no significant clinical factors associated with the response" is worded incorrectly. Change to "Although no clinical factors associated significantly with the response"
3. I disagree with the authors statement that "further studies on BTK inhibition with the R2A regimen for the GCB regimen are needed" as the CR rate here is 18%, implying that 2 of the 11 GCB patients had a CR - not worth further study in this population.
4. The treatment "CAR-T" is spelled incorrectly and should be referred to as "CAR T-cell therapy".
5. The citation 14 is attached to the sentence about axi-cel, but appears to be a study of genetics in NEJM. Double check references.
6. The citation of 8 claiming CAR T-cell 'real world' shows less benefit is disingenuous as there are multiple other high profile publications showing the opposite that the author do not cite.
7. The authors claim the R2A regimen is an alternative to CAR T-cell therapy - this is a wild overstatement. CAR T-cell therapy has an OS advantage over ASCT in 2L therapy, and this trial shows a relatively short PFS in a different population of patients. Suggest the authors reconsider how bold they choose to be with comparing this regimen with others.
- 8.

RESPONSE TO REVIEWERS' COMMENTS

Reviewer #1 (Remarks to the Author):

Park et al. present data of a phase 2 study of the BTK inhibitor acalabrutinib in combination with rituximab and lenalidomide in patients with relapsed/refractory diffuse large B-cell lymphoma (DLBCL). The authors report promising efficacy and overall a manageable toxicity profile. Furthermore, the authors report response rates in molecularly defined subgroups of DLBCL.

Response

We thank you for the constructive comments. The response to your criticism is provided below each comment. We feel that our manuscript has been improved, guided by your comments, and we hope that our revisions are satisfactory.

Major criticism:

1. How many patients received prior CAR T-cell therapies?

Response

Due to regulatory and reimbursement issues in the Republic of Korea at the time when the patients were enrolled (July 2019 ~ Jan 2021), none of the patients received prior CAR T-cell therapy. For clarification, we added the sentences as follows:

- In the RESULTS section: None of the patients had previously received prior CAR T-cell therapy, which was due to regulatory and reimbursement issues in the Republic of Korea at the time of the trial.
- In the DISCUSSION section: Fourth, all patients in this study did not receive CAR T-cell therapy prior to R2A. Therefore, the efficacy of R2A in patients who received prior CAR T-cell therapy cannot be evaluated in this study.

2. How many patients had lymphomas with double or triple hit constellation?

Response

There were 35 patients with information on the rearrangement results. Among them, one had triple-hit diffuse large B-cell lymphoma, achieving a complete response (CR) with a progression-free survival (PFS) of 32 months. The patient's disease did not progress until the data cutoff. We added this information as follows:

- In the RESULTS section: Three patients had double-hit and one patient had triple-hit DLBCL. One of three patient with double-hit DLBCL responded, and the PFS of the three patients were 0.5, 1.4 and 3.3 months, respectively. The one patient with triple-hit DLBCL achieved complete remission (CR) with a progression-free survival (PFS) of 32 months. The patient did not experience disease progression until the data cutoff.

- In Table 1, the following rows were added:

Double/triple hit – no. (%)	
Yes	4 (6.1)
No	31 (47.0)
Not available	31 (47.0)

3. In their NGS analyses which genes were investigated?

Response

We performed targeted gene NGS, and the detailed gene list is provided in Supplementary Method 3. Please refer to Supplementary Method 3 for the specific gene information.

4. How was the response in unclassifiable DLBCL according to the Lymphgene subtypes?

Response

Thank you for your comment. The response rate among unclassifiable DLBCL patients with available genomic data was 57.5% (23 out of 40 patients). Unfortunately, due to our use of targeted gene NGS, most tumor samples with available genomic data were classified as unclassifiable DLBCL by Lymphgen subtypes. Only two patients, classified as MCD subtype, were placed into one of the Lymphgen subtypes. Both patients with the MCD subtype responded to the R2A regimen.

In an effort to explore other strategies for classification and understand genomic correlations with response, we incorporated details on the R2A regimen efficacy based on the LymphPlex classification. LymphPlex employs a simplified 38-gene algorithm to classify DLBCL into subtypes, utilizing similar terminology as the Lymphgen classification. (Reference: Shen R, et al. Simplified algorithm for genetic subtyping in diffuse large B-cell lymphoma. Signal Transduct Target Ther. 2023;8.) This alternative classification method successfully categorized nine samples into specific subtypes. We have presented the results below. We hope this additional information would be satisfactory.

- In the **RESULT** section: A total of two patients were classified into MCD subtype by LymphGen classification, and the remaining 40 patients were unclassifiable. Total of nine patients were classified into specific subtype by LymphPlex classification (Four EZB-like-MYC⁻, three TP53^{Mut}, one BN2-like and one MCD-like subtype). The outcomes of patients according to these subtypes are summarized in **Table 3**. Notably, three patients classified into subtypes that are known to have NF-κB activation (MCD, MCD-like, and BN2-like subtype) responded to the R2A regimen. There was no difference according to each subtype classification (**Supplementary Figure 4**).

- In the **DISCUSSION** section: Due to the limitation of the targeted sequencing, most

of the samples could not be classified into specific subtypes as proposed by previous literature. However, we observed that subtypes with NF- κ B activation, which is related to B cell receptor signaling and BTK pathway, responded to the R2A regimen. Since only a few samples were evaluated in this study, conducting further research using whole-exome sequencing to assess the association of the LymphGen classification with the R2A regimen would be valuable. This is particularly important as the combination regimen is likely to be effective in specific subgroups of lymphoma with an enrichment in the BTK pathway.

- In the **METHODS** section: For classification of tumors according to genetic status, we used algorithms proposed by LymphGen classification and LymphPlex, which allow classification similar to LymphGen classification with simplified 38-gene algorithm.

- Figure 3 was revised as follows:

Figure 3

- Supplementary Figure 4 was revised as follows:

Supplementary Figure 4

5. The authors should perform gene expression profiling to determine ABC and GCB DLBCL and correlate these findings to response and PFS, OS, etc.

Response

Thank you very much for your suggestion. In response, we conducted additional RNA sequencing analysis using available samples to identify ABC and GCB DLBCL subtypes and correlate these findings with the efficacies, as guided by your comment.

- In the **METHODS** section: Cell-of-origin subtyping using RNA sequencing was conducted on tumor samples for which whole transcriptome sequencing data were available. We followed the procedures described in previous literature.

- In the **RESULTS** section: A total of 30 patients had available tissue for RNA sequencing, and expression profiling for cell-of-origin subtyping was conducted in these patients. Among them, 13, 11, and 6 patients were classified into the activated B-cell, germinal center B-cell, and unclassifiable subtypes, respectively. The ORR was 69.2% (95% CI 41.3 – 88.8), 54.5% (95% CI 26.0 – 80.1), and 83.3% (95% CI 41.0 – 99.2) in each of the activated B-cell, germinal center B-cell, and unclassifiable subtype, respectively, with no significant differences ($p = 0.613$). There were no significant differences in PFS and OS according to the cell-of-origin subtype (**Supplementary Figure 5**).

- Supplementary Figure 5 was revised as follows:

6. How was the assay determining BTK expression established. Which controls were used?

Response

We used the cell extract from TMD8 cell as positive control. As for the assay determining BTK expression from FFPE slides, we followed a rigorous protocol that involved multiple steps. Firstly, we deparaffinized the slides using xylene and sequentially rehydrated them with 100%, 95%, and 70% ethanol, followed by 18MOhm water. Next, we collected the rehydrated specimens in AFA tubes and resuspended them in an extraction buffer consisting of 1% SDS and 50mM Tris-HCl (pH=8.0). The specimens were homogenized for 6 minutes using Covaris M220, followed by a 1-hour incubation of the AFA tubes at 99°C and 750rpm in an Eppendorf thermomixer for antigen retrieval and protein extraction. We then transferred the resulting extracts to Eppendorf tubes and centrifuged them at 15000g to collect the supernatants, which we diluted 10-fold with a dilution buffer consisting of 1% TX100, 50mM Tris-HCl (pH=8.0), and 150mM NaCl to reduce the SDS concentration. Finally, we measured BTK expression levels using BTK antibodies (D3H5 and D6T2C, Cell Signaling Technology) and single-molecule fluorescence imaging, calibrated the results with the area of the FFPE specimens, and used a threshold of 10000 to distinguish between BTK high and BTK low levels. These details on the assay determining BTK expression are provided in the Supplementary Method 2.

Reviewer #2 (Remarks to the Author):

Response

We thank you for the constructive comments. The response to your criticism is provided below each comment. We feel that our manuscript has been improved, guided by your comments, and we hope that our revisions are satisfactory.

- multiple of the non-significant results were over-interpreted. Interpreting as a wide confidence interval and large p-value as indicated that a group "tended to" do anything feels like it doesn't recognize the limited precision available for many of the results presented. Presenting the results and confidence intervals without additional narrative description would be appropriate.

Response

Thank you for your criticism. We acknowledge that we may have overemphasized several non-significant results, potentially misleading readers. In the revised version, we have made adjustments in the RESULTS section. Non-significant results are now presented without additional narrative description. For p-values between 0.05 and 0.1, we explicitly state that the observed difference is not statistically significant. We hope these revisions address your concerns.

- The ORR in patients with non-GCB type DLBCL was 61.7% (95% CI 46.8 – 74.8) and that in patients with GCB type DLBCL was 36.4% (95% CI 13.5 – 66.8). The ORR tended to be higher in patients with non-GCB type DLBCL (61.7%, 95% CI 46.8 – 74.8) compared to GCB type DLBCL (36.4%, 95% CI 13.5 – 66.8), although the difference was not statistically significant ($p = 0.179$).

- Among patients with Bcl-2 positive IHC, those with Myc positive IHC tended to have a lower ORR compared with the patients with Myc negative IHC, but the difference was not statistically significant (35.3% [95% CI 16.6 – 59.4] vs. 68.4% [95% CI 44.5 – 85.3], $p = 0.093$).

- Among patients who were Bcl-2 IHC positive, those with Myc positive IHC had a median PFS of 3.9 months (95% CI 1.5 – 24.0), while those with Myc negative IHC had a median PFS of 5.1 months (95% CI 1.5 – 24.0); however, the difference was not statistically significant ($p = 0.29$, **Supplementary Figure 2**). tended to have shorter median PFS compared to other patients (median PFS 3.9 months [95% CI 1.5 – 24.0] vs. 5.1 months [95% CI 3.5 – 15.7], $p = 0.29$, **Supplementary Figure 2**).

- Patients with high BTK expression on single-molecule fluorescence imaging tended to have a longer PFS (median PFS 5.2 months [95% CI, 4.5 – NA]) than those with low BTK expression, but the difference was not statistically significant (median PFS 2.0 months [95% CI, 0.8 – NA]; $p = 0.055$; **Supplementary Figure 6**).

- given the very small N in biomarker subgroups the conclusion in the abstract biomarker enrichment designs are warranted seems like an over-statement.

Response

Thank you for your criticism. We agree that the results related to the biomarker should be interpreted as hypothesis generation, and asserting the need for biomarker enrichment designs may have been overstated. While we anticipate that some of our findings may be validated in subsequent clinical trials involving glofitamab, poseltinib, and lenalidomide, it would be premature to emphasize the need for a biomarker enrichment design. In response to your comments, we have revised the conclusion in the abstract as follows:

- In the ABSTRACT section: In conclusion, the R2A regimen showed significant efficacy in patients with aggressive R/R B-cell NHL, ~~warranting further study with biomarker enrichment~~ with exploratory biomarkers showing potential associations.

Reviewer #3 (Remarks to the Author):

Dr. Koh and colleagues have conducted an important clinical trial evaluating the combination of rituximab, lenalidomide, and acalabrutinib. The rationale of the combination is strong, and the trial was well designed.

Response

We appreciate your constructive comments, and our responses to each comment are provided below. We believe that our manuscript has been enhanced with the guidance from your feedback, and we hope that our revisions meet your expectations.

Introduction:

1. The first sentence is confusing as it does not define where CR rates are 76% - with first line therapy?

Response

Thank you for your comment. We clarified the sentence as below:

- In the INTRODUCTION section: Since the advent of Rituximab, treatment for CD20 positive aggressive B cell non-Hodgkin's lymphoma (NHL) represented by diffuse large B cell lymphoma (DLBCL) has significantly advanced, achieving complete remission (CR) for as high as 76% of patients with first-line rituximab combined chemoimmunotherapy.

2. The "significant portion" with refractory disease is vague, please define as significant could be interpreted broadly.

Response

Thank you for your comment. We agree that “significant portion” is a vague term. We clarified the sentence as below:

- In the INTRODUCTION section: Moreover, ~~significant proportion~~ around 15% of patients are reported to be refractory to commonly used first line rituximab-based regimens.

3. Language - it is difficult to define "an ideal regimen" for R/R B-cell NHL broadly as R2A. This language should be toned down.

Response

Thank you for your comment. We agree with your suggestion and have revised the sentence as follows:

- In the INTRODUCTION section: Theoretically, the combination of a BTK inhibitor with rituximab and lenalidomide may be an ideal effective regimen for R/R B-cell NHL.

4. The references cited here are unusual. Citing one paper about a prospective CAR T_cell trial but not others, then citing a review papers about commercial CAR T-cell vs other therapies but not a review paper about DLBCL in general. Suggest closer attention to which papers are cited here.

Response

Thank you for your thorough evaluation of our manuscript. We acknowledge that CAR T-cell therapy is considered one of the optimal treatment choices for relapsed/refractory DLBCL patients. Despite its effectiveness, financial toxicity and reimbursement challenges persist. In Republic of Korea, Tisagenlecleucel, a CAR T-cell therapy, was only recently approved and reimbursed in 2022, with strict indications controlled by government and a high cost. In this challenging landscape, we recognized the necessity to explore alternative approaches for these patients. Consequently, we conducted a clinical trial investigating the combination of acalabrutinib, rituximab, and lenalidomide. To ensure readers comprehend the context, we have revised both the Introduction and Discussion sections as follows:

- In the INTRODUCTION section: Although the development of therapeutics such as chimeric antigen receptor (CAR) T-cell (CAR-T) therapy and T-cell engaging antibody has successfully treated some of relapsed/refractory (R/R) DLBCL patients who has grave prognosis, the challenges posed by the high costs and reimbursement issues associated with these therapies have prompted the exploration of alternative approaches.

- In the DISCUSSION section: However, the application of CAR T-cell therapies may be hindered by high costs and reimbursement issues. In Republic of Korea, tisagenlecleucel was only recently approved for relapsed/refractory DLBCL patients in 2022, yet ongoing financial and administrative challenges persist. Furthermore,

the promising results of CAR T-cell therapies come at the expense of severe toxicities, including cytokine storms and encephalopathy, which pose potential fatal risks and demand intensive management. Therefore, in situations where CAR T-cell therapies are not available, the R2A regimen may be considered for relapsed/refractory DLBCL patients. In addition, the real-world data on CAR T-cell therapy showed far less PFS benefit (median PFS 5.2 months), suggesting alternate therapies may be as efficacious as CAR T-cell therapy in select clinical scenarios. Considering the efficacies and relatively easily manageable toxicities of the R2A regimen, the regimen can be an attractive alternative to CAR T-cell therapy for patients with R/R B-cell NHL, and even for patients who failed to CAR T-cell therapies as these often target CD19 while R2A regimen targets include CD20.

Materials and Methods

1. The authors mention patients "failed at least one line of treatment". This language is not acceptable - the patient did not fail the treatment, but instead the treatment failed the patient. We should not blame our patients for treatment failure. Change to some version of disease relapsed or refractory to at least 1 line of treatment.

Response

Thank you for your thoughtful comment. We totally agree with your comment and we changed the sentence as below:

- In the METHODS section: "Patients should also have failed experienced relapse or refractory to at least one line of treatment if they were ineligible for autologous stem cell transplantation and at least two lines of treatment if they were candidates for autologous stem cell transplantation."

2. Presumably second line CAR T-cell therapy was not available as it is not mentioned in the exclusion for patients who had only 1 prior therapy (but ASCT is). Correct?

Response

Your comment is right. Due to regulatory and reimbursement issues in Republic of Korea at the time when the patients were enrolled (July 2019 ~ Jan 2021), CAR T-cell therapy was practically unavailable in Republic of Korea. For clarification, we added the sentence mentioning this as below:

- In the RESULTS section: the sentence "None of patient had previously received prior CAR T-cell therapy, which was due to the regulatory and reimbursement issues in Republic of Korea at the time of trial." was added.

- In the DISCUSSION section: Fourth, all patients in this study did not receive CAR T-cell therapy prior to R2A. Therefore, the efficacy of R2A in patients who received prior CAR T-cell therapy cannot be evaluated in this study.

3. Why are there no details about the treatment doses and schedule in the manuscript,

but there are details about a safety cohort?

Response

Thank you for your comment. We initially thought that it may be redundant to mention the treatment disease and schedule in the manuscript and Supplementary Method 1. However, we agree that it would be more appropriate to provide brief details on the treatment doses and schedule in the main manuscript. We added regarding these as follows:

- In the METHODS section: the sentences “A treatment cycle consisted of days 1–28, during which rituximab 375 mg/m² was administered intravenously on day 1, lenalidomide 20 mg orally once daily from days 1 to 21, and acalabrutinib 100 mg orally twice daily from days 1 to 28. The cycle was repeated every four weeks for up to six cycles if the disease did not progress during this period.” were added.

4. The definition of "pathology" is not sufficient. Were assays to determine cell of origin or RNA sequencing conducted? If so, which assays?

Response

Thank you for your comment. Regarding the pathologic diagnosis and findings, we did not mandate any specific assays but collected the formal reports of pathologists from each center. As RNA sequencing to determine the lymphoma subtype is not reimbursed in Republic of Korea, RNA sequencing is not routinely performed in clinical practice and all pathologic diagnosis were based on the morphologies and immunohistochemistry findings. We clarified this within the sentence as below:

- In the METHODS section: ~~Pathology~~ Pathologic reports by board-certified pathologist in each center were collected. The reports consisted of pathologic diagnosis, immunohistochemistry (IHC) reports and fluorescence in situ hybridization reports, which include ~~and immunohistochemistry (IHC) reports,~~ including B-cell lymphoma 2 (Bcl-2), B-cell lymphoma 6 (Bcl-6), Myc, and Ki-67 protein expressions, ~~translocations of Bcl-2, Bcl-6, and Myc were obtained from the medical records of each center.~~ The assays in each patient were performed as per the clinical decisions from each center.

In the meantime, we performed RNA sequencing analysis using available samples to identify ABC and GCB DLBCL subtypes and correlate these findings with the efficacies for exploratory biomarker analysis purpose as follows:

- In the METHODS section: Cell-of-origin subtyping using RNA sequencing was conducted on tumor samples for which whole transcriptome sequencing data were available. We followed the procedures described in previous literature.

- In the RESULTS section: A total of 30 patients had available tissue for RNA sequencing, and expression profiling for cell-of-origin subtyping was conducted in these patients. Among them, 13, 11, and 6 patients were classified into the activated B-cell, germinal center B-cell, and unclassifiable subtypes, respectively. The ORR

was 69.2% (95% CI 41.3 – 88.8), 54.5% (95% CI 26.0 – 80.1), and 83.3% (95% CI 41.0 – 99.2) in each of the activated B-cell, germinal center B-cell, and unclassifiable subtype, respectively, with no significant differences ($p = 0.613$). There were no significant differences in PFS and OS according to the cell-of-origin subtype (**Supplementary Figure 5**).

- Supplementary Figure 5 was revised as follows:

Supplementary Figure 5

5. Was CNS involvement excluded?

Response

CNS involvement was not excluded and there were two patients who had primary CNS lymphoma. We added the comments on the CNS involvement as clarification for potential readers.

- In the METHODS section: the sentence “Patients with central nervous system involvement could also be enrolled.” was added.

Results

1. BCL6 expression is mentioned prior to Double expressor - please confirm the authors are not including BCL6 expressing via IHC as a part of "double expressor", as this term only applies to MYC and BCL2 protein expression.

Response

Thank you for your comment. We confirm that we refer to “double expressor” as DLBCL with both Bcl-2 and Myc overexpression. We clarified this in the Results section as below:

- In the RESULTS section: ~~Seventeen patients had a double-expressor phenotype of DLBCL.~~ There were 17 patients having a double-expressor phenotype (both Bcl-2 and Myc overexpression) of DLBCL.

2. Were any patients double hit with MYC and BC2 or BC6 translocations?

Response

Out of the 35 patients with available information on rearrangement results, one was diagnosed with triple-hit diffuse large B-cell lymphoma. This patient achieved a CR with a PFS of 32 months, and there was no progression in the disease until the data cutoff. We have included this information in the findings section below:

- In the RESULTS section: Three patients had double-hit and one patient had triple-hit DLBCL. One of three patient with double-hit DLBCL responded, and the PFS of the three patients were 0.5, 1.4 and 3.3 months, respectively. The one patient with triple-hit DLBCL achieved complete remission (CR) with a progression-free survival (PFS) of 32 months. The patient did not experience disease progression until the data cutoff.

- In Table 1, the following rows were added:

Double/triple hit – no. (%)	
Yes	4 (6.1)
No	31 (47.0)
Not available	31 (47.0)

3. "The enrolled patients received lenalidomide 20mg" - this needs to be clarified with an English editor - I presume this means that all subsequent patients received 20mg from cycle 1.

Response

Thank you for your comment. We were meant to mention exactly as you commented. We changed the sentence as below:

- In the RESULTS section: ~~The enrolled patients~~ The patients subsequently enrolled received lenalidomide 20 mg daily starting from the first cycle.

4. One patient with PR received acalabrutinib maintenance, why not the other patients with PR? Why did 2 patients with CR not receive the acalabrutinib maintenance?

Response

Thank you for your comment. According to the protocol, patients who responded to the R2A regimen, defined as partial response (PR) or complete response (CR), and completed the six cycles of the combination were offered maintenance treatment with acalabrutinib monotherapy for up to 1 year. The patient who achieved a PR and received acalabrutinib did not experience disease progression after completing the six cycles, leading to the continuation of maintenance treatment. In contrast, the two patients who achieved CR but did not receive acalabrutinib maintenance experienced disease progression before completing the six cycles. For clarification, we have added a description of the criteria guiding the indication for patients who receive maintenance acalabrutinib, as outlined below:

- In the **METHODS** section: Those who responded to the R2A regimen, defined as partial response (PR) or complete response (CR), **and completed the six cycles of the combination were offered to receive** maintenance treatment with acalabrutinib monotherapy up to 1 year.

5. It is difficult to understand how only 33% of patients had any hematologic toxicity. This rate seems very low - were the other 66% of patients completely free from cytopenias?

Response

Thank you for your comment. We acknowledge the significant concerns regarding toxicity, particularly hematologic toxicity. Notably, our combination regimen does not include any cytotoxic chemotherapeutic agents, which suggests a potentially lower impact on bone marrow function. In addition, acalabrutinib is not commonly associated with hematologic toxicity and lenalidomide dose was 20 mg daily, which is lower than the dose used in other ibrutinib-combination clinical trials (Radhakrishnan Ramchandren, et al., eClinicalMedicine, 2022, DOI <https://doi.org/10.1016/j.eclinm.2022.101779>; Andre Goy, et al., Blood, 2019, DOI <https://doi.org/10.1182/blood.2018891598>). Two previous clinical trials on ibrutinib, rituximab, lenalidomide combination for relapsed/refractory DLBCL reported the hematologic toxicities in 20% ~ 44% of patients (Radhakrishnan Ramchandren, et al., eClinicalMedicine, 2022, DOI <https://doi.org/10.1016/j.eclinm.2022.101779>; Andre Goy, et al., Blood, 2019, DOI <https://doi.org/10.1182/blood.2018891598>). One clinical trial of acalabrutinib, rituximab, lenalidomide combination for mantle cell lymphoma reported hematologic toxicity up to 38% of patients (Jia Ruan, et al, ASH 2022 abstract, DOI <https://doi.org/10.1182/blood-2022-158656>). Considering these references, the observed 33% incidence of hematologic toxicity in our study appears to be within the expected range. A review of clinical records during follow-up revealed that, at least at the time of CBC evaluation, the remaining 66% of patients did not experience cytopenias. While it would have been valuable to assess potential mid-cycle cytopenias, the low incidence of febrile neutropenia (only 2 patients) suggests the regimen may be safe concerning hematologic toxicity. We have included this information on the incidence of hematologic toxicity from previous clinical trials in the Discussion section as follows:

- In the **DISCUSSION** section: In the entire study cohort, one-third of the patients experienced hematologic toxicities, **which is consistent with previous clinical trials. The reported incidences of the hematologic toxicities in two previous clinical trials on ibrutinib, rituximab, and lenalidomide combination for R/R DLBCL reported incidence ranging from 20% to 44%.(11,23) Another clinical trial on R2A for mantle cell lymphoma reported hematologic toxicity in up to 38% of patients.(24) reported a previous real-world study that evaluated the efficacy of a similar regimen as the first-line treatment of elderly DLBCL patients.**

6. It is difficult to understand how none of the patients had any change in quality of life on the three metrics mentioned. Even patients with lymphoma that was refractory to treatment had no change to quality of life? Even the patient that came off due to the eosinophilia? This raises questions of how many patients had pre and post therapy assessment of QoL.

Response

Thank you for your comment. Owing to the low compliance of patients with lengthy questionnaires, we were only able to include responses from 28 patients for the analysis of changes from baseline to cycle 2, and 36 patients for the analysis of changes from baseline to end-of-trial. Our intention was to demonstrate that there were no significant changes in quality of life (QoL) that might suggest a substantial adverse impact of the R2A regimen on QoL. However, in response to your feedback, we recognize that readers may be interested in understanding how QoL changes according to treatment response. To address this, we conducted analyses based on treatment response categories (CR/PR or SD/PD). The results indicated that, overall, there were no statistically significant changes in QoL during the R2A regimen, except for global health status scores, which showed a significant increase from baseline to end-of-trial in responders, and for functional scores, which showed a significant decrease from baseline to end-of-trial in non-responders. To enhance clarity, we annotated the response categories using colors in the figures displaying QoL and provided the corresponding p-values according to response groups, as detailed below.

As for the patient who exhibited eosinophilia experienced a decrease in functional score as well as global health status score, along with an increase in symptom scales score. Unfortunately, this patient did not provide an end-of-trial QoL questionnaire response.

- In the RESULTS section: There were 28 patients with responses available for analyses on the changes from baseline to cycle 2. No significant changes in functional score, global health score, or symptom scale were observed at cycle 2. Responses available for analyses on the changes from baseline to end-of-trial were obtained from 36 patients. No significant changes in the scores were observed in the trial cohort, except for global health score which increased significantly ($p = 0.013$). When the patients were divided into responders and non-responders, patients who responded showed significant increase in global health score ($p = 0.044$), whereas patients who did not respond showed significant decrease in functional score ($p = 0.034$, **Supplementary Figure 3**). ~~None of the patients experienced deterioration in the quality of life in terms of the functional score, global health score, or symptom scale (Supplementary Figure 3).~~

- In the DISCUSSION section: In addition, the quality of life of these patients was generally acceptable, although responses to treatment seemed to have impact and there may be a selection bias towards patients who responded to the survey.

- **Supplementary Figure 3** was revised as below:

Supplementary Figure 3

7. The sentence "all EZB-related gene-mutated and ST2-related gene wild-type patients (N = 5)

293 responded to R2A, whereas none of the wild-type and ST2-related gene-mutated patients" is worded incorrectly. What is the second use of wild type to mean - EZB-related wild type? I have not seen these lymphGen subsets described as mutated or wild type in other manuscripts. It may be more clear what the authors are implying to say genes associated with EZB or ST2, and not use the label of EZB-wildtype or ST2-mutated.

Response

Thank you for your constructive feedback. This analysis was undertaken due to our interest in understanding the response of patients based on LymphGen subtypes. Unfortunately, our targeted gene NGS approach resulted in the classification of most tumor samples into the unclassifiable DLBCL category by LymphGen subtypes, with only two patients classified into the MCD subtype. In response to your comments, we recognize that our initial approach of analyzing outcomes based on mutations in genes related to each subtype may be confusing to readers and susceptible to the multiple comparisons fallacy. Consequently, we have decided to present the outcomes of patients according to specific classifications in a more descriptive manner.

In addition to the LymphGen classification, we explored alternative strategies for classifying DLBCL patients to identify genomic correlations with response. We have now included a description of the R2A regimen efficacy according to LymphPlex classification, which employs a simplified 38-gene algorithm to classify DLBCL into

subtypes using similar terminology as LymphGen classification. This approach successfully classified nine samples into specific subtypes, and we have presented the results below with revised discussions. We hope this description would be satisfactory.

- In the **RESULT** section: A total of two patients were classified into MCD subtype by LymphGen classification, and the remaining 40 patients were unclassifiable. Total of nine patients were classified into specific subtype by LymphPlex classification (Four EZB-like-MYC⁻, three TP53^{Mut}, one BN2-like and one MCD-like subtype). The outcomes of patients according to these subtypes are summarized in **Table 3**. Notably, three patients classified into subtypes that are known to have NF-κB activation (MCD, MCD-like, and BN2-like subtype) responded to the R2A regimen. There was no difference according to each subtype classification (**Supplementary Figure 4**).

- In the **DISCUSSION** section: Due to the limitation of the targeted sequencing, most of the samples could not be classified into specific subtypes as proposed by previous literature. However, we observed that subtypes with NF-κB activation, which is related to B cell receptor signaling and BTK pathway, responded to the R2A regimen. Since only a few samples were evaluated in this study, conducting further research using whole-exome sequencing to assess the association of the LymphGen classification with the R2A regimen would be valuable. This is particularly important as the combination regimen is likely to be effective in specific subgroups of lymphoma with an enrichment in the BTK pathway.

- In the **METHODS** section: For classification of tumors according to genetic status, we used algorithms proposed by LymphGen classification and LymphPlex, which allow classification similar to LymphGen classification with simplified 38-gene algorithm.

- Figure 3 was revised as follows:

Figure 3

- Supplementary Figure 4 was revised as follows:

Supplementary Figure 4

8. "LymphGene" is mis-spelled

Response

Thank you for your comment. We have corrected the misspelling to 'LymphGen' throughout the manuscript.

9. Comparing mutations between the subsets as the authors are doing (WT here,

mutation there) is not standard. Typically patients are assigned to one subtype, and not classified based upon wild type status in other subtypes. This is concerning for multiple comparisons fallacy where random combinations of subsets could appear significant. This analysis is deeply flawed, and should not be presented as though the "Subset" of EZB-WT/ST2-WT is favorable. Which subtype are those patients? It is acceptable to show that EZB subtype or ST2 subtype have outcomes, but to lump together all other subtypes and call them effectively "not EZB or ST2" is not appropriate.

Response

Thank you for your criticism. We take your comments very importantly regarding the analysis. In response, we have revised the analysis to description on the outcome according to LymphGen classification and LymphPlex classification from previous literatures. Please refer to the response to your comment number 7 regarding RESULTS section.

Discussion

1. The authors compare their results with other studies in other subtypes of B NHL, when this trial was nearly all DLBCL. These comparisons are flawed due to different types of NHL having differential responses, but the authors present them as comparable.

Response

We appreciate your observation, and we acknowledge that comparing the efficacy results in different tumor populations is inherently flawed. Consequently, we opted to compare our results with the historical findings of trials on relapsed/refractory DLBCL (or other aggressive B-cell NHL). All the literature cited in the first paragraph of the Discussion section focused on trials for relapsed/refractory DLBCL. While one study [Wang M, et al., Leukemia, 2013, DOI: <https://doi.org/10.1038/leu.2013.95>] included other aggressive B-cell NHL cases, the majority of enrolled patients were diagnosed with DLBCL. To elucidate our approach, we thoroughly discussed and compared these results in the Discussion section.

It is important to note that these comparisons are indirect, and we explicitly stated in the same paragraph that "direct comparisons between this trial and other trials should not be interpreted as our results being superior." Additionally, for clarity, we briefly mentioned in the text that our study cohort predominantly comprised DLBCL, while some referenced studies in the Discussion section encompassed DLBCL as well.

- In the DISCUSSION section: In this phase II clinical trial on patients with R/R B-cell NHL, mostly DLBCL, a combination of acalabrutinib with rituximab and lenalidomide was effective, with manageable toxicities and a fair quality of life. The ORR of the trial regimen (54.5%) was higher than that of BTK inhibitor acalabrutinib monotherapy (24%) for R/R B-cell NHL DLBCL reported by Strati et al.(24) and that

of ibrutinib therapy (35%) reported by Graf et al.,(25) with. In addition, the ORR was also higher than that in the previous literature on the rituximab/lenalidomide combination; 33% in Wang et al.,(26) and 35% in Zinzani et al.(27)

2. The sentence "Although no significant clinical factors associated with the response" is worded incorrectly. Change to "Although no clinical factors associated significantly with the response"

Response

Thank you for your comment. We changed the sentence as you commented.

3. I disagree with the authors statement that "further studies on BTK inhibition with the R2A regimen for the GCB regimen are needed" as the CR rate here is 18%, implying that 2 of the 11 GCB patients had a CR - not worth further study in this population.

Response

Thank you for your feedback. We acknowledge that it would be inappropriate to solely apply our regimen to the GCB subtype due to the low response rate. However, we observed two patients with GCB-type DLBCL who experienced complete remission (CR) with the R2A regimen. In the sentence you referenced, our intention was to suggest that there may be a small subset of GCB-type DLBCL that could respond to BTK inhibition. We recognize that the original sentence could be misinterpreted as indicating significant efficacy for GCB-type DLBCL. Therefore, we have revised the sentences below to ensure that readers do not overestimate the findings of our study.

- In the DISCUSSION section: However, ~~for some~~ **two** patients with GCB-type DLBCL ~~their best objective response is~~ **experienced** CR. As the combination of rituximab and lenalidomide may enhance tumor susceptibility to acalabrutinib, further **preclinical studies to find subset of GCB-type DLBCL that may be dependent on the BTK signaling may be needed.** ~~on the BTK inhibition using the R2A regimen for GCB-type DLBCL are needed.~~

4. The treatment "CAR-T" is spelled incorrectly and should be referred to as "CAR T-cell therapy".

Response

Thank you for your comment. We changed the abbreviation to "CAR T-cell therapy" throughout the manuscript.

5. The citation 14 is attached to the sentence about axi-cel, but appears to be a study of genetics in NEJM. Double check references.

Response

Thank you for your thorough evaluation of our manuscript. We double-checked the reference and changed to the appropriate literature on axicabtagene ciloleucel.

6. The citation of 8 claiming CAR T-cell 'real world' shows less benefit is disingenuous as there are multiple other high profile publications showing the opposite that the author do not cite.

Response

Thank you for your criticism. We reviewed our sentences in the Discussion section, and totally agree with your comment. We removed the parts where we mentioned on the 'real world' data.

7. The authors claim the R2A regimen is an alternative to CAR T-cell therapy - this is a wild overstatement. CAR T-cell therapy has an OS advantage over ASCT in 2L therapy, and this trial shows a relatively short PFS in a different population of patients. Suggest the authors reconsider how bold they choose to be with comparing this regimen with others.

Response

Thank you for your criticism. We agree that CAR T-cell therapy must be considered in R/R DLBCL patients, and the above statement is not appropriate to be claimed. Our intention was to emphasize that the R2A regimen could be considered in clinical settings where CAR T-cell therapies are not available, possibly due to financial constraints, reimbursement issues, or concerns about toxicity. We have revised the relevant sections regarding the comparison with CAR T-cell regimens to ensure that potential readers of our manuscript do not misinterpret our findings.

- In the DISCUSSION section: However, the application of CAR T-cell therapies may be hindered by high costs and reimbursement issues.(7) In Republic of Korea, tisagenlecleucel was only recently approved for R/R DLBCL patients in 2022, yet financial and administrative challenges persist.(32) Furthermore, the promising results of CAR T-cell therapies come at the expense of severe toxicities, including cytokine storms and encephalopathy, which pose potential fatal risks and demand intensive management.(25,26) Therefore, in situations where CAR T-cell therapies are not available, the R2A regimen may be considered for R/R DLBCL patients. Recently, several BiTEs have shown promising efficacies. Based on the results of R2A, a combination of a BTK inhibitor and a CD20-targeting BiTE may be effective in overcoming the potential limitations of CAR T-cell therapy and failures to CD19-targeting CAR T-cell therapy. We are currently conducting a clinical trial to evaluate the combination of glofitamab, poseltinib, and lenalidomide for patients with R/R DLBCL (ClinicalTrials.gov identifier: NCT05335018) In addition, the real-world data on ~~CAR T-cell~~ therapy showed far less PFS benefit (median PFS 5.2 months), suggesting alternate therapies may be as efficacious as ~~CAR T-cell~~ therapy in select clinical scenarios. Considering the efficacies and relatively easily manageable

toxicities of the R2A regimen, the regimen can be an attractive alternative to ~~CAR-T-cell~~ therapy for patients with R/R B-cell NHL, and even for patients who failed to ~~CAR-T-cell~~ therapies as these often target CD19 while R2A regimen targets include CD20.

REVIEWERS' COMMENTS

Reviewer #1 (Remarks to the Author):

The authors have adequately addressed the raised concerns.

Reviewer #2 (Remarks to the Author):

Thank you for the comprehensive response. Two minor edits that are needed:

Line 172: Delete the "however," the p-value is far from significant and the absolute difference is modest so there does not seem to be a discrepancy even descriptively.

Line 254: May be more clear to write "the difference did not reach statistical significance" because the $p=0.0055$.

Reviewer #3 (Remarks to the Author):

The manuscript is much improved with the changes made.

Two further observations

1. There is no category called "triple hit". These patients have double hit lymphoma and the additional genomic anomaly is not additive or distinct, and therefore is irrelevant. Suggest use correct terminology of either HGBCL with MYC and BCL2 or BCL6 or double hit.
2. The Supp Fig 4 has an error in the legend where TP53 and unclassified are reversed in the number at risk for the right hand panel which presumably is for OS.

RESPONSE TO REVIEWERS' COMMENTS

Reviewer #1 (Remarks to the Author):

The authors have adequately addressed the raised concerns.

Response

We thank you very much for the constructive comments.

Reviewer #2 (Remarks to the Author):

Thank you for the comprehensive response. Two minor edits that are needed:

Response

We thank you very much for the constructive comments.

Line 172: Delete the "however," the p-value is far from significant and the absolute difference is modest so there does not seem to be a discrepancy even descriptively.

Response

Thank you for your thorough review. We deleted the "however" at Line 172. - In the RESULTS section: while those with Myc negative IHC had a median PFS of 5.1 months (95% CI 1.5 – 24.0); however, the difference was not statistically significant
--

Line 254: May be more clear to write "the difference did not reach statistical significance" because the $p=0.0055$.

Response

Thank you for your thorough review. We changed the phrase as you mentioned at Line 254. - In the RESULTS section: but the difference was not statistically significant did not reach statistical significance
--

Reviewer #3 (Remarks to the Author):

The manuscript is much improved with the changes made.

Response

We thank you very much for the constructive comments.

Two further observations

1. There is no category called "triple hit". These patients have double hit lymphoma and the additional genomic anomaly is not additive or distinct, and therefore is irrelevant. Suggest use correct terminology of either HGBCL with MYC and BCL2 or BCL6 or double hit.

Response

Thank you for your thorough review. We changed the term as you mentioned.

- In the RESULTS section: ~~Three patients had double-hit and one patient had triple-hit DLBCL. One of three patient with double-hit DLBCL responded, and the PFS of the three patients were 0.5, 1.4 and 3.3 months, respectively. The one patient with triple-hit DLBCL achieved complete remission (CR) with a progression-free survival (PFS) of 32 months. The patient did not experience disease progression until the data cutoff.~~ Four patients had double-hit DLBCL, with one patient having all of MYC, BCL2, and BCL6 rearrangements. The one patient with all of the three rearrangements achieved complete remission (CR) with a progression-free survival (PFS) of 32 months. The patient did not experience disease progression until the data cutoff. In the remaining three patients, one patient responded, and the PFS of the three patients were 0.5, 1.4 and 3.3 month.

2. The Supp Fig 4 has an error in the legend where TP53 and unclassified are reversed in the number at risk for the right hand panel which presumably is for OS.

Response

Thank you very much for your important comments. We changed the error in the Supp Fig 4 as you mentioned as follows:

Supplementary Figure 4